# The Overlooked and the Overstudied: A Scoping Review of Qualitative Research on Pursuing Sexual, Romantic, and Loving Relationships Through Online Dating

**DOI:** 10.3390/bs15030247

**Published:** 2025-02-21

**Authors:** Plata S. Diesen, Lene Pettersen, Faltin Karlsen

**Affiliations:** School of Communication, Leadership and Marketing, Kristiania University of Applied Sciences, 0153 Oslo, Norway; lene.pettersen@kristiania.no (L.P.); faltin.karlsen@kristiania.no (F.K.)

**Keywords:** dating apps, online dating, online relationships, romantic relationships, scoping review, qualitative research

## Abstract

This paper presents a scoping review of the qualitative research (N = 125) on the use of online dating sites and applications for adults pursuing relationships, including sex, love, and romance, from 2014 to 2023. Our review supports previous literature reviews’ findings, which reveal that research on the topic is predominantly focused on young, well-educated, ethnic-majority, and primarily female heterosexuals or men seeking men in Western societies. Hence, a sample-selection bias shapes our scientific understanding of online dating, leaving other user groups underrepresented. Despite the diversity of scientific fields involved in qualitative research, the methods used are notably similar, indicating a relatively narrow scope in both demographic variables and research approaches. Although the researched themes and perspectives appear diverse at first glance, the research often centers on problem-oriented topics, such as the risks and emotional aspects of online dating, insecurities in self-presentation, negative technological communication traits, and the de-romanticization of society. We conclude that, despite the growing body of research on online dating, significant areas of the topic remain unexplored. There is a need for broader, more inclusive research to fully understand the complexities of online dating in the digital age.

## 1. Introduction

The proportion of single individuals has been rising over recent decades in industrialized countries ([65]). Meeting others through dating applications (apps) and websites has become a common practice among those seeking casual interactions, sexual encounters, or long-term relationships ([40]; [54]). Computer-based matchmaking technologies first emerged in the 1960s, and one of the earliest online dating websites, Kiss.com, was founded in 1994. The Match Group, Inc. quickly became a leading global provider of dating products, with brands such as Tinder, Match, OkCupid, Hinge, Pairs, and PlentyOfFish, among others ([37]). In 2009, Grindr introduced the first geosocial app for gay men, and pioneered location-based dating platforms ([49]). Dating apps are mobile software platforms designed to facilitate connections between individuals ([40]), while online dating websites serve a similar purpose via computer-based platforms ([40]). Since both mediums aim to facilitate interactions with strangers for diverse personal motivations, we will refer to both collectively as online dating in this article.

Today, online dating has been normalized in many cultures and regions, and it plays a significant role in how individuals meet others in the 21st century. Currently, more than 384.15 million people worldwide use dating apps ([63]). While Tinder remains the most popular dating app globally ([63]), numerous other apps cater to diverse communities based on religious, political, cultural, gendered, or sexual orientations (e.g., lesbian, gay, bisexual, transgender, LGBTQ+), as well as niche markets such as individuals with disabilities, hyper-local communities, and other specific user groups. Moreover, while they are popular among young adults, older demographics are increasingly engaging with these platforms, including those in their 40s, 50s and beyond, with 38% of Tinder’s user base aged 35 and older ([11]). Many apps cater to different age groups, with some specifically targeting older adults looking for serious relationships or casual connections.

In parallel with technological and societal shifts, an increasing amount of research has been conducted from a variety of academic disciplines on various aspects of dating apps, their usage, and their users. This body of research is significant for several reasons. First, it sheds light on topics of critical societal relevance in an era where many individuals are involuntarily single or childless ([29]). Knowledge of how people choose partners and form romantic relationships is important for a deeper understanding of human behavior. Being in a healthy relationship is often associated with improved life satisfaction, mental health, and well-being ([33]; [51]), although [3] ([3]) suggest that this picture is more complex and nuanced. Second, findings from research on online dating address a universal topic of public interest, as love and relationships concern nearly everyone. Consequently, dating research findings influence media discourses and public understandings of relationships and online dating practices. [32] ([32]) find that dating and dating app experts in the Norwegian public media landscape consequently portray online dating as a high-risk activity, balancing strategic rules and human needs. These experts describe dating apps as virtual playgrounds or games where people act recklessly to achieve their goals, making dating difficult and leading to involuntary singleness. Through various popular media channels with large audiences, dating experts and their expertise gain a form of authority to define what constitutes “good” and “bad” dating, identify problems, and suggest better dating practices ([32]).

Given the importance of online dating, literature reviews play a crucial role in summarizing prior research to identify gaps and propose directions for future studies ([45]; [74]; [35]). Previous literature reviews on online dating have focused on quantitative research ([31]), problematic usage patterns ([9]), sociodemographic and psychosocial correlates ([12]), regional practices in Asia ([36]), gay dating app studies ([80]), and approaches from media and communication perspectives ([79]). Moreover, many of these previous literature reviews and samples seem to approach the topic from perspectives rooted in psychology, mental health, and human behavior. As emphasized by [59] ([59]), it is crucial to identify the subjects and populations that have been investigated to gain a comprehensive understanding of the research landscape. However, to date, no comprehensive literature review has been conducted on the extensive body of qualitative research addressing online dating across academic disciplines. This scoping review aims to fill that gap by examining the demographic characteristics in qualitative research on online dating within the qualitative research, as well as to identify the recurring themes and perspectives. Furthermore, considering the diverse methodologies employed in qualitative research ([4]), this study also seeks to elucidate the methods that have been utilized. Accordingly, this article addresses the following research questions (RQs):What demographic aspects are explored in qualitative research on online dating, and what are the prevalent research methods?What key themes and perspectives are identified in qualitative research on online dating?

In terms of demographic aspects, we decided to include age, sexual identity, gender, ethnicity, and educational level to enhance the overall understanding provided by previous literature reviews. Our decision to keep RQ1 unified stems from the recognition that who is being studied and how they are being studied are deeply interconnected. By examining both demographics and research methods together, we gain a more comprehensive understanding of potential gaps, biases, and trends in qualitative research on online dating. This approach allows us to assess not only which populations are over- or underrepresented but also how methodological choices may shape the knowledge being produced.

## 2. Previous Literature Reviews of Online Dating

Key findings from previous literature reviews provide an important background for our study. [31] ([31]) included 72 quantitative studies in their systematic review, while [9] ([9]) reviewed 43 studies, 82% of which were quantitative. Similarly, [12] ([12]) reviewed 70 studies, with 80% being quantitative. [36] ([36]) examined 19 studies, incorporating both qualitative and quantitative research. [80] ([80]) and [79] ([79]) did not specify the number of studies included in their review sample. Our scoping review consists of 125 articles, almost twice as many as any of the abovementioned, as scoping reviews tend to include a much larger number of studies ([42]).

[31]’s ([31]) systematic review is especially interesting for our study, as they examine the quantitative body of research, while we look at the qualitative research. Although the sample of research reviewed are not listed in [31]’s ([31]) article, the majority of their article’s references are publications emphasizing technological and psychological aspects in online dating. Our review of qualitative research may both support and complement their findings, as well as those of other literature reviews, helping to identify hidden knowledge gaps critical for future studies.

One of the key findings in [31]’s ([31]) review revealed a strong focus on young adults in this literature, with an average participant age of 25.9 years. This is also a tendency in [12]’s ([12]) review. They reported that psychosocial studies predominantly targeted individuals aged 18–30. Similarly, [36] ([36]) reported that the majority of dating app users in Asia were young, educated, and open about their sexual orientation ([36]). Nevertheless, [31] ([31]) found that studies covering a broader age range suggest that motivations for using dating apps shift as users age.

When directing our lens to the distribution of gender and sexual orientation in quantitative research, [31] ([31]) find that gender was generally balanced and addressed as a binary concept in their sample of the literature. Regarding sexual orientation, among the 72 studies reviewed by [31] ([31]), 31 focused on heterosexual participants, 21 examined men who have sex with men (MSM), and 20 did not report sexual identity. Psychosocial studies have often focused on MSM, highlighting significant diversity in age and geographical context, encompassing both rural and urban areas ([12]). Similarly, [36] ([36]) observed this pattern in Asian studies.

Asian studies further explored challenges like risky sexual behavior, substance use, and harassment, particularly among women and MSM ([36]). These studies also emphasized safety measures and mate preferences, with education emerging as a significant factor, and [36] ([36]) noted that a bachelor’s degree was often considered the minimum standard. [9] ([9]) highlighted personality traits such as sociability, sensation-seeking, sexual permissiveness, and anxious attachment as predictors of greater usage in their review on problematic user patterns, and correlated problematic use with motives like self-esteem enhancement and sex-seeking. In addition, it has been noted that the facilitation of casual encounters raises public health concerns, such as risks of sexually transmitted infections and substance use ([12]; [9]).

[80]’s ([80]) review of gay dating apps focused on themes such as online self-presentation, the evolution of gay communities, and interpersonal relationships in digital spaces. [79] ([79]) extended this analysis, when they highlighted how dating apps both shape and are being shaped by social forces. They advocated for increased research on underexplored socio-cultural and relational contexts and stressed the importance of well-contextualized studies. Nevertheless, as studies often fail to build upon one another and thus contribute to a unified understanding across the field, [12] ([12]) and [31] ([31]) find that existing research is characterized by a lack of coherence and integration.

As we noted in the introduction, previous literature reviews and samples seem to be approached from perspectives in psychology, mental health, and human behavior. [31]’s ([31]) review finds that quantitative research has mainly focused on dispositional factors such as gender and psychological traits, and that the outcomes of online dating research have predominantly centered on sexual behaviors ([31]). Quantitative surveys on online dating primarily investigate measurable user behaviors, such as app usage frequency, swiping patterns, and motivations such as entertainment, socializing, and romantic relationships ([31]). They also find that there is a notable lack of validated measurement instruments and scales to assess users’ motivations and behaviors in online dating.

In terms of the samples of participants used in quantitative studies, [31] ([31]) find that the majority were identified as white/Caucasian, and that research overwhelmingly originated from the Global North, particularly the United States, Western Europe, and Australia. Overall, the Global South remains underrepresented, even when Spanish-language studies are considered ([12]). The research often uses non–representative samples, with young and well–educated individuals, primarily from Western, industrialized, wealthy, and democratic societies ([31]).

## 3. Methodology

As an umbrella term, qualitative research comprises a variety of methods, aiming to provide in-depth insights into people’s subjective experiences ([4]). Qualitative investigations into social media—a topic that is analogous to dating apps ([50])—have frequently employed interviews and focus groups as primary data collection methods, and are often complemented by quantitative survey data ([61]; [48]). As previously noted, although several literature reviews on dating apps have been conducted, none have yet reviewed the qualitative research on users of online dating sites and apps.

To explore the social phenomenon of online dating and address the research questions guiding this review, we conducted a scoping review focusing on peer-reviewed qualitative studies that examine the use of online dating platforms for pursuing relationships, including those centered on sex, love, and romance. According to [45] ([45]), scoping reviews are effective for examining complex and diverse bodies of literature, particularly when the aim is to systematically map existing research, identify knowledge gaps, or analyze key concepts in a field. This stands in contrast to systematic reviews, which synthesize empirical evidence from a set of studies to answer specific questions, often focused on effectiveness or qualitative evidence. Scoping reviews are widely employed in social sciences for literature synthesis, but [35] ([35]) have highlighted inconsistencies in reporting search strategies. They advocated the adoption of tools such as PRISMA and engaging professional librarians.

The literature searches and selection process were carried out in three stages. First, the primary author conducted pilot searches over a period of approximately two months, from April to June 2023. To mitigate potential strategic and methodological difficulties that are associated with database searches ([45]; [35]), we collaborated with an experienced librarian from our research institution at this stage. Second, all three authors performed concurrent searches simultaneously in August and September 2023, to reduce the chance of errors and biases ([45]), using an identical search strategy to that in stage one. Finally, all three authors conducted new searches in early January 2024 to ensure the inclusion of publications from the entire year of 2023. To ensure transparency and replicability of the evidence sources, we present a comprehensive description of the data extraction process ([45]; [35]).

### 3.1. Search Strategy

The databases selected for our search included Web of Science, PsycINFO, Communication & Mass Media Complete, and Medline. We chose Web of Science and Communication & Mass Media Complete for their extensive coverage in social and media sciences. Additionally, PsycINFO and Medline were included due to their specialization in medical and psychological research, thereby ensuring a comprehensive exploration of perspectives, including those related to online dating in mental health, quality of life, and disability research. We also examined the reference lists of all included articles to identify further relevant studies. Our searches were restricted to original, peer-reviewed journal articles in the English language, addressing qualitative research on adults’ (>18 years) use of dating apps and online dating in searching for sex, love, and romance. Mixed-method articles, which combine qualitative research with surveys, were also included. We chose a full ten-year time range, from 2014 to 2023.

### 3.2. Search Words

Search words were categorized into three main groups: methodological terms, dating terms, and internet terms, which ensured that at least one search term from each category was included. Table 1 presents the search categories and terms included.

We applied the Boolean operator AND to combine terms across different categories and the Boolean operator OR to combine search words within a single category. We opted to distinguish between “app” and “apps” due to the use of “app*”, which resulted in approximately 4000 additional hits, including unrelated search words such as “approach”. Additionally, we combined the search term “dating” with relevant terms to avoid conflicting meanings, such as “date trees” or “time dating”.

In PsycINFO, field one (“Methodological terms”) was set to “Abstracts”. Fields two and three (“Dating terms” and “Internet terms”) were set to “Text word”, due to an insufficient number of results when restricted to abstracts alone. Furthermore, we restricted the searches to include only peer-reviewed articles from 2014 to 2023 that provided abstracts and were in English. In Medline, we set field one to “Abstracts”, and fields two and three to “Text word”. Medline does not provide the peer-reviewed setting, as all articles in the Medline database are peer-reviewed. Therefore, we limited the search to include articles from 2014 to 2023 that provided abstracts and were in the English language. In Web of Science, field one was set to “Abstract”, and fields two and three were set to “All fields”, which is equivalent to “Text word” in PsycINFO and Medline. However, it is important to note that Web of Science lacks a distinct filter to restrict searches to peer-reviewed articles. Although most journals provided by Web of Science are peer-reviewed, the database does not maintain a comprehensive list of the peer-review status, which constitutes a limitation in our searches. Finally, we limited the search to include articles (document type) in the English language from 2014 to 2023. In Communication & Mass Media Complete, we set field one to “AB Abstract or Author-Supplied Abstract”, which is equivalent to “Abstract” in the previous three databases. Fields two and three were set to “TX All Text”, which is equivalent to “Text word” and “All fields”. We limited the searches to include peer-reviewed articles (document type) in academic journals (publication type) from January 2014 to August 2023, in the English language.

### 3.3. Exclusion Categories

Initially, our exclusion criteria were limited to articles based solely on quantitative methods involving adolescents (<18 years) and articles focused on sexually transmitted diseases. However, searches revealed a significant number of articles that, although addressing or mentioning online dating, did not primarily address adult individuals seeking love, romance, or sexual relationships online. Consequently, we excluded these records from our review. To enhance transparency and better align with our initial aim, we established more specific exclusion categories, as detailed in Table 2.

### 3.4. Prisma Flow Chart

Figure 1 presents a Prisma flow chart ([41]) that illustrates the stages of article identification and selection, as meticulously reviewed, and cross-checked by authors one, two, and three. Our initial searches across multiple databases yielded a total of 532 articles for screening. After removing 167 duplicates using the Endnote automation tool, an additional 30 duplicates were identified through manual checks, resulting in a total of 335 articles. We reviewed the remaining abstracts and excluded a total of 181 abstracts based on the exclusion categories (Table 2), which resulted in 154 records for full-text reading. During the full-text reading, we excluded another 36 articles based on the established exclusion criteria. Further, we added seven articles by reference-list reading. As a result, we have included 125 articles in our review, which are listed in Appendix A.

### 3.5. Data Analysis

As qualitative research is context–specific, synthesizing results in a literature review poses challenges ([75]). Nevertheless, insights from multiple qualitative studies can be consolidated, and several approaches to synthesize qualitative data have been proposed ([6]). In this review, we employed thematic synthesis ([69]) alongside descriptive statistical analysis of study demographics and methods using Excel. The coding process involved a close reading of the articles, identifying key themes within the introduction, results, and discussion sections. Through axial coding, these themes were grouped into broader categories, which were then further refined through selective coding into the four main themes discussed in this article.

## 4. Results

As a starting point, one early observation was that the number of articles steadily increased with time, rising from three publications in 2014 to 34 in 2023. Upon initial examination, we encountered a landscape that appeared to be complex and somewhat perplexing. However, although authors frequently emphasized the distinctiveness of their work, many studies seemed to converge on similar characteristics. Our results are structured in line with the two research questions asked in this article.

### 4.1. Results Part One

This part addresses research question 1: What demographic aspects are explored in qualitative research on online dating, and what are the prevalent research methods?

#### 4.1.1. Age

Due to inconsistent reporting of mean age across studies—many of which only categorized participants into generational cohorts without providing specific age data—we found it necessary to apply the same categorization to all available data. This revealed a pronounced bias toward younger age groups. Generation Z, born between 1997 and 2012, was featured in 98 of the 125 articles and Millennials, born between 1981 and 1996, were represented in 89 articles. When studies included Generation X (1965–1980), Boomers (1946–1964), and the Silent Generation (1928–1945), the mean age consistently skewed younger. Researchers suggest that age differences on dating apps may be more impactful than gender differences, with older adults seeking stable relationships, and younger individuals pursuing casual encounters ([52]) and displaying more fluid sexual identities ([19]). However, we found that the reviewed articles encompassing a wider age range often overlooked generational differences.

#### 4.1.2. Gender and Sexual Identity

Heterosexuals received the largest focus in our reviewed articles. However, a closer look at the heterosexual samples revealed a significant gender imbalance: 28 articles focused on female-majority samples and 12 on female-only samples, compared to only four articles focusing on male-majority samples and two on male-only samples. When comparing genders, researchers suggested that men were often more direct and inclined towards casual encounters, whereas women tended to seek long-term relationships ([57]; [26]).

Although some studies had included a range of sexual orientations, heterosexuals remained the predominant focus, with little attention given to how sexual minorities use these platforms. Within the LGBTQ+ community, the distribution of research revealed notable disparities: 31 studies focused on men who have sex with men (MSM), 4 on women who have sex with women (WSW), and 7 on other LGBTQ+ identities. Research on MSM predominantly emphasized brief sexual encounters or “hook-ups”, with comparatively limited attention to MSM seeking deeper, romantic relationships.

#### 4.1.3. Educational Levels

Information on participants’ educational backgrounds was included in 51 articles, which revealed a clear predominance of higher education levels. Many of the studies that neglected to specify educational backgrounds still indicated connections to higher educational environments, such as having conducted the interviews near university campuses. Among the studies that had included participants with lower educational backgrounds, the majority still had higher education credentials. When studies incorporated multiple educational levels, these differences were often not addressed in the results.

#### 4.1.4. Origin

Regarding the countries of origin represented in our sampled articles, 26 out of the world’s 195 countries were featured. Additionally, three studies broadly identified participants’ origins as Asia, South America, or Europe. In total, 38 articles originated from the USA, followed by the UK with 19 articles, China and Australia with 13 each, and Canada with 12 articles. Some studies included participants from multiple countries. Grouping by continent revealed the largest clusters in Europe, with 46 publications, followed by North America with 40, Asia with 26, Australia with 15, Africa with 2, and South America with 1 article.

#### 4.1.5. Ethnicity and Minorities

Ethnicity was addressed in 57 articles, most of which were from the USA. Among these studies, although diverse ethnicities were generally included in the samples, five studies specifically excluded Caucasians. However, the majority of participants were frequently white/Caucasian. European and Asian authors tended to focus on participants’ nationality and the language used in interviews, which created an impression of a predominantly ethnic-majority sample. Additionally, there is a significant gap in the representation of individuals with disabilities, with two articles addressing the topic.

#### 4.1.6. Qualitative Research Methods

In the reviewed material, 107 studies relied predominantly on interviews as their primary or sole method. Focus groups were utilized in fifteen studies, dating profile analysis in nine studies, and other methods such as ethnography, autoethnography, research diaries, and walk-through methods were also employed. Although our review identified some multimethodological approaches, they were not predominant. The number of participants in interview studies varied widely, ranging from 5 to 249, with a mean of 31, while studies using focus groups did not always specify the number of participants per group but reported using between 2 and 13 focus groups, with a mean of 4.

### 4.2. Results Part Two

This part will present findings that address research question 2: What key themes and perspectives are identified in qualitative research on online dating?

#### 4.2.1. Problem-Oriented Perspectives

Upon reviewing the sampled articles, we initially noted that the research predominantly adopted problem-oriented perspectives. Success stories related to dating apps, especially regarding their potential of facilitating users finding lasting love, were seldom emphasized. Instead, the focus was primarily on the challenges faced by dating app users and negative societal trends, with the theoretical frameworks generally offering a critical view of modern relationship dynamics.

Although this is representative of the overall impression, it is crucial to acknowledge that the studies also presented a diverse array of perspectives. They spanned various scientific disciplines, and many utilized complex theoretical frameworks relevant to their specific fields. Nonetheless, certain themes were particularly prominent within the material. In our thematic analysis, we have synthesized the prevalent theories and perspectives into four key sub-themes: the de-romanticization of love and intimacy; risks and emotional vulnerabilities; self-presentation; and media and technology. The themes ranged from large societal and cultural perspectives to topics revolving around individual users and the role of specific technology. In the following sections, we will present each of the fours themes.

#### 4.2.2. De-Romanticization of Love and Intimacy

The studies we reviewed often present a dystopian view, framing online dating as emblematic of troubling changes in how people pursue love and relationships. This perspective suggests that a traditional romantic relationship, with its clear expectations of commitment and intimacy, is increasingly being eroded by the inherent commodification in dating technology ([7]; [29]).

In our sample of studies, dating apps are described as creating a landscape where traditional ideals are increasingly overshadowed by data-driven algorithms and market-driven logics ([5]). Along the same lines, [13] ([13]) concludes that users are ambivalent when it comes to forming meaningful relationships, and relates this to an over-abundance of connections, a process that is deeply intertwined with the neoliberal market and individual consumption habits. [15] ([15]) finds that users struggle to balance cultural expectations of finding true love with the belief that dating apps were the only viable option, and argue that the commercialization and rationalization that have come to define the interdependency between technology and intimacy create an existential burden for them.

From this perspective, online dating represents a disturbing shift towards the dehumanization of relationships by translating traditional courtship into data-driven systems, which often prioritize efficiency over emotional depth. It describes a tendency for casual encounters and long-term relationships to converge, entangling the pursuit of “true love” with fleeting, impersonal interactions ([16]; [39]). Implicit in this perspective, however, is the notion that traditional courtship and marriage is morally superior to the current practices ([20]), which might overshadow the potential progress and opportunities which are brought about by online dating.

#### 4.2.3. Risks and Emotional Vulnerabilities

In our sample of articles, a myriad of emotional, social, and safety challenges related to online dating are described. However, the reports are ambivalent and vary between groups. Ghosting, for example, the act of ending communication with a potential partner without explanation and avoiding contact thereafter ([60]), is described as causing negative emotional impacts for some, and to be negligible to others ([60]; [70]).

A recurring sub-theme is that interacting with quasi-strangers involves significant risks ([5]), but the challenges manifest differently across various user demographics. For heterosexual women, for example, online dating is generally described as fraught with risks and emotional strains, where they may navigate a hostile online environment where harassment, deception, and risks of violence are constant concerns ([47]; [26]; [27]). Heterosexual men, on their side, try to avoid rejection and humiliation, and struggle with self-image ([55]; [17]). Dating apps are also often framed as an arena that facilitates the reinforcement of negative stereotypes ([72]; [2]; [71]). The fact that white heterosexuals frequently choose partners of their own color is, for instance, described as concealing and normalizing racism, rationalized as personal choices ([44]), and consequently as excluding people of color ([27]).

Research on LGBTQ+ individuals, which comprise a substantial part of the material, are associated with other types of emotional vulnerabilities, with dating apps reportedly failing to deliver the empowerment they promise. The pervasive concerns about public health issues might erode trust and safety within the online dating landscape ([1]), and overall, LGBTQ+ individuals encounter distinct challenges related to identity and identifiability, objectification, privacy, and societal stigma. Queer young women, for example, have been found to frequently experience frustration and disempowerment ([22]), while older generations fear stigma ([67]). Bisexual women have reported experience of marginalization and hypersexualization ([25]). Transgender and non-binary individuals, meanwhile, are concerned with safety risks and fetishization ([24]).

Our review also reveals that the consequences that men might face from being identified as same-sex-attracted varied depending on culture and society. In India, for example, where homosexuality was illegal until 2018, the threat of social shaming and significant violence is described as prevalent ([8]; [58]; [46]).

#### 4.2.4. Self-Presentation

As users of dating apps and online dating sites present themselves to others through their profiles, it is not surprising that much of the existing research has focused on self-presentation, drawing on [23]’s ([23]) concept of impression management. A consistent finding across the literature is that users aim to present an ideal, yet authentic, version of themselves ([77]; [57]; [76]). However, despite having spent two decades with social media, it is notable how many studies highlight persistent feelings of insecurity regarding self-presentation and interpreting others’ communications, which leads back to the above section on emotional vulnerability.

Several articles describe users as engaging in self-presentation akin to self-branding or personalized advertisements, reflecting a marketized view of love ([5]). This involves managing impressions to fit certain stereotypes, with a frequent emphasis on youthfulness ([58]). However, the type of self-presentation varies according to user demographics. Heterosexual men, for instance, are generally found to prioritize physical appearance and traditional masculinity, aiming to appear fit without seeming vain ([76]), whilst older adults emphasize personality, interests, and values in their profiles ([78]). For transgender and non-binary individuals, self-presentation is closely linked to expressions of gender identity ([24]). Disabled users, on the other hand, were associated with self-presentations where overcoming isolation and accessibility barriers was a common narrative ([38]).

#### 4.2.5. Media Technology

The fourth and last sub-theme comprises studies that examine affordances and algorithms in media technology. Affordances are defined as actual or perceived properties that enable or constrain use ([53]), whilst algorithms involve pairing massive data sets through coding, organizing, and extracting data ([10]). This sub-theme focuses on structural elements of the apps, such as swiping technology and geo-localization, as well as the entertainment aspects of dating apps. A recurring observation is that people often have expectations of how algorithms work that are not met when using the apps. For example, users might use dating apps to increase diversity in their encounters but end up liking people who are similar to them ([43]). Conversely, studies describe how geo-localization in dating apps has changed how marginalized individuals meet, such as those belonging to the MSM group, especially in non-Western societies. Consequently, the apps provide opportunities that were previously unavailable, though not without risks ([8]; [62]). Once again, we encounter the issue of marginalization, as affordances and algorithms could reinforce racialized preferences ([2]) and promote heteronormativity ([14]).

## 5. Discussion and Conclusions

Directed by our first research question—What demographic aspects are explored in qualitative research on online dating, and what are the prevalent research methods?—we structured our findings into age, gender and sexual identity, educational level, origin, ethnicity and minorities, and qualitative research methods.

A key finding concerns the research sample’s age. Consistent with findings from other literature reviews of online dating ([12]; [31]; [36]), we found that the research on online dating suffers from a sample-selection bias as most of the studies focus on individuals in their twenties, typically students. Challenges with relying on samples or databases consisting mainly of students, and the weaknesses related to the generalizability of the theoretical conclusions reached in these studies, is an old and well-known problem ([56]). However, researchers who publish in social psychology’s major journals still rely heavily on student samples ([28]). Young students’ practices with online dating and use of dating apps are logically not representative of a larger population, nor universal. Despite the significant presence of younger users on dating apps ([11]), it is necessary to recognize that online dating services cater to a much wider age range ([64]). Neglecting the perspectives and experiences of older users risks producing an incomplete and not fully representative understanding of contemporary dating dynamics. Future research should include a wider range of life stages as this would provide valuable insights into how age and different life experiences intersect with technology use, romantic goals, and relationship formation in digital spaces.

Another key finding is that heterosexuals dominate the samples. The heterosexual gender imbalance in qualitative studies contrasts with findings from quantitative research, which indicated a slightly higher proportion of male participants ([31]). Moreover, the predominance of female participants in qualitative research contrasts with app statistics, which show that heterosexual men are overrepresented among dating app users ([11]), suggesting that existing qualitative research disproportionately emphasizes women’s experiences. Furthermore, while dating apps are recognized for their significance in fostering connections within the gay community ([80]), a notable lack of balanced representation for women who are seeking women (WSW) and other LGBTQ+ groups remain. Addressing these disparities is vital for fostering a more inclusive understanding of dating dynamics and even for advancing feminist approaches that prioritize gender equality and its implications for dating practices.

When examining the sample’s educational levels, we found a clear predominance of individuals with higher education. By primarily examining the experiences of well-educated dating app users, we risk overlooking the perspectives of a broader demographic, which is crucial for a nuanced understanding of diverse user groups. In their review on Asian studies, [36] ([36]) highlighted how higher educational levels shape behaviors and preferences, with a bachelor’s degree often considered as a minimum. We argue that this underscores the need to explore the perspectives of less–educated users to provide a more inclusive understanding of online dating.

In accordance with broader research trends, studies on online dating exhibit a pronounced Western-centric bias, which has resulted in a cycle of research that focuses primarily on Western contexts ([81]). [36] ([36]) highlighted the individualistic nature of Western cultures, in contrast to the holistic orientation of Asian cultures, suggesting notable differences in their use of online dating. It is imperative to acknowledge and address cultural variations, and to include diverse ethnic-minority perspectives. Addressing and rectifying these biases is crucial for a more comprehensive and nuanced understanding of the diverse facets of online dating. Expanding the scope to incorporate the experiences of ethnic minorities could yield valuable insights and significantly enrich research perspectives.

Regarding the various methodological tools employed in qualitative research, we found that 107 of our sample’s 125 articles used interviews as their primary or sole method. Given that qualitative methods are inherently multimethodological, interpretative, and naturalistic in their approach ([18]; [4]), the significant reliance on interviews may limit the exploration of the phenomenon. Although the predominance of interviews in the research landscape is not problematic in itself, the over-reliance on one single methodological approach, together with a narrow demographic scope, can potentially affect our comprehensive understanding of the research subject. Interviews are indeed a prevalent choice in qualitative research ([68]; [48]), however, they possess inherent limitations. These include reliance on self-reported data, potential issues with selective and limited memory, constraints due to short interview durations, contextual limitations, influence from the participant’s desire to present themselves in a particular way, and the impact of the interviewer’s presence and context ([73]; [68]). Exploring alternative methodologies, such as netnographical studies, analyses of in-app communication, and investigations into how dating transpires across other platforms or life domains, could offer a broader or different perspective on the phenomenon.

To conclude on our first research question, our review highlights a disproportionate focus on young, white, heterosexual individuals with higher education from Western countries, with studies on these groups often constrained by a limited range of methods. Meanwhile, ethnic minorities, individuals with disabilities, those from lower educational backgrounds, individuals in mid-life and beyond, those in varied life stages, women seeking women (WSW), and other LGBTQ+ groups remain understudied. Surprisingly, despite their significant presence on dating platforms, heterosexual men have also been comparatively overlooked. The limited scope of methods used in the literature further narrows the overall perspective, as these conventional approaches fail to capture the diversity of experiences across these underrepresented groups, reinforcing an incomplete view of online dating’s impact.

Future research should address the cultural bias in existing studies by focusing more on these overlooked demographics, as highlighted in our review. This bias is particularly problematic when research from diverse regions relies on such studies to interpret local phenomena. In qualitative research, cultural nuances and interpretive frameworks play a critical role in shaping understanding ([4]). For example, while love is often seen as universal, cultural variations influence how romantic expressions are understood and enacted ([30]). Similarly, dating experiences differ significantly based on age, identity, and cultural context, making it essential to consider how these factors shape individual experiences and expectations. Recognizing these differences is key to gaining a comprehensive understanding of online dating dynamics.

When examining our second research question—What key themes and perspectives are identified in qualitative research on online dating?—our analysis revealed a dominant focus on problem-oriented perspectives. Here, we recognize that the problem-oriented approach may indeed reflect challenges in today’s dating arenas. However, the focus may also, in part, stem from the relative novelty of online dating. Historically, new media technologies have always raised concerns about effects on individuals or groups, with mental health, risk behavior, and social norms being recurring topics ([66]). When—or if—media technology becomes fully integrated into society, concerns usually subside and shift toward newer media technologies. In this capacity, the problem-oriented focus may reflect concerns that are not as prevalent in contemporary society as our ten-year span of articles might suggest.

We found that success stories, such as those where dating apps lead to lasting relationships, marriages, or long-term partnerships, are frequently downplayed or omitted. By focusing predominantly on themes such as risks, emotional vulnerabilities, and the commodification of relationships, the literature obscures the opportunities dating apps provide for people seeking to connect in ways that may not have been possible through traditional avenues. For individuals who may struggle to meet partners through conventional means—due to limited social circles, unique relationship goals, or specific demographic characteristics—online dating can offer a valuable alternative ([21]). These platforms provide new possibilities for connecting with like-minded individuals and expanding the dating pool beyond traditional boundaries of geography and social circles. For those who find it difficult to meet potential partners offline, whether because of personality traits or niche communities, online dating can offer a unique space for fostering meaningful connections that might otherwise remain out of reach. Moreover, by consistently focusing on the same demographics, such as middle-class students from majority groups, we miss the opportunity to capture these diverse experiences of individuals from underrepresented communities, thereby restricting our understanding of the broader impact of online dating.

It is also worth considering whether the problem-oriented framing in the academic literature extends into the broader public discourse surrounding online dating. In popular media, dating experts often portray online dating as a high-risk activity, casting dating apps as virtual “games” where individuals recklessly pursue fleeting connections. This narrative reinforces the idea of online dating as chaotic and superficial, where short-term goals lead to failed relationships or involuntary singleness ([32]). This public portrayal mirrors the academic focus on risks, perpetuating negative stereotypes about online dating and potentially discouraging individuals from embracing its potential for meaningful connections. Given the strong influence of media portrayals on public perception, it is essential that the discourse surrounding online dating evolves to recognize both its limitations and its capacity to foster relationships that might otherwise be difficult to form. Only by incorporating a more balanced perspective—one that addresses both the pitfalls and the possibilities—can we gain a clearer understanding of the true impact of online dating on modern relationships.

## 6. Limitations

This literature review is not without limitations. Similar to other literature reviews, the choice of databases employed in our study comes with certain considerations. For instance, one of the databases used for our searches (Web of Science) lacks a comprehensive list of the peer-review status of its articles. Although the majority of their journals are peer-reviewed, and we meticulously examined each included article individually, this aspect could introduce a limitation to our study. Additionally, the limitations extend to our selected search terms, specific types of literature (e.g., journal articles), and language (English only), which may have led to the omission of relevant content ([61]; [34]), with the language restriction potentially contributing to the overrepresentation of the Global North. Although we could have expanded our search to include a larger number of databases and alternative sets of search terms, we chose a focused approach to preserve the clarity and feasibility of the study.

We recognize that, despite that the three authors of this review are from three different social science fields, our own academic backgrounds, individual perspectives, and limitations may have influenced our focus and interpretation. The studies under review span a diverse array of scientific disciplines, each of which employs intricate theoretical frameworks that are relevant to their respective domains. Nevertheless, by focusing on themes rather than specific fields, we believe we have provided a cohesive analysis that transcends disciplinary boundaries. This thematic approach allows for a broader synthesis of insights, ultimately offering a more comprehensive understanding of the subject matter.

## Figures and Tables

**Figure 1 behavsci-15-00247-f001:**
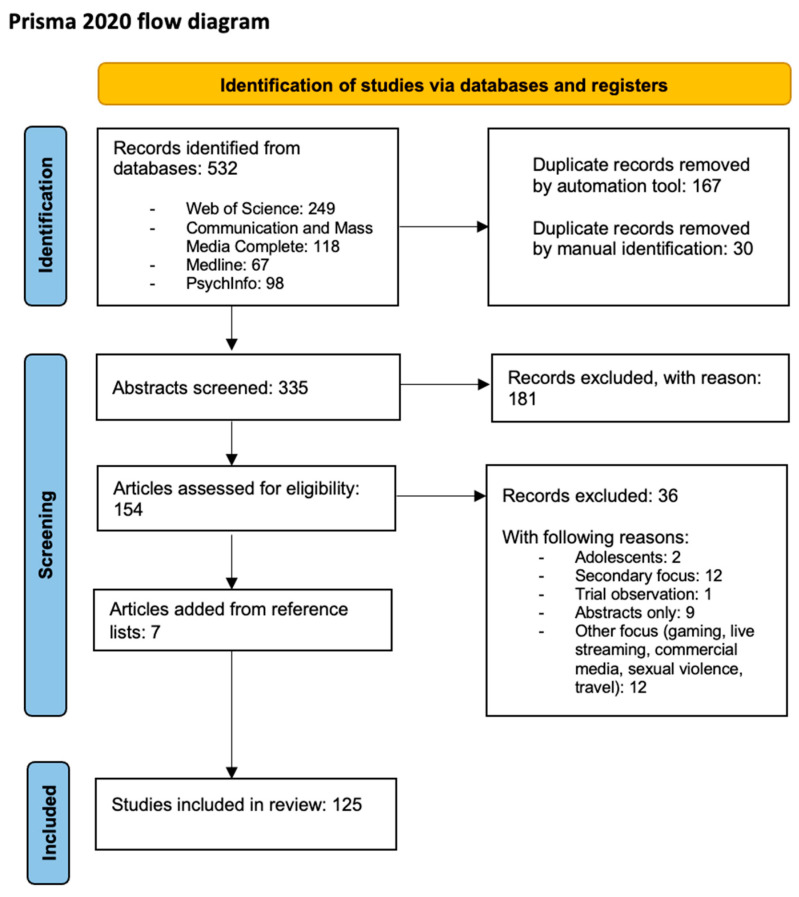
Prisma flow diagram of identified and selected articles.

**Table 1 behavsci-15-00247-t001:** Search categories and terms.

Search Category	Search Terms
Methodological Terms	Qualitative OR interview* OR ethnograph* OR autoethnograph* OR “mixed method*” OR “focus group*”
Dating Terms	“dating app” OR “dating apps” OR “dating application*” OR “online dating” OR repartnering OR “internet dating” OR “single parent* dating” OR “dating platform*”
Internet Terms	online OR internet OR app OR apps OR application*

**Table 2 behavsci-15-00247-t002:** Overview of exclusion criteria and themes.

Exclusion Categories	Themes
Medical	HIV, STDs, brain damage, mental disorders, diagnostics.
Business generating	Porn, prostitution, catfishing, scams, drug sales, marketing and campaigning, movie industry.
Advisories	Parenting advice, sex education, health professional’s perspectives.
Other technological platforms	Social media, gaming, live streaming, workplace romance, AI, and Big Data.

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
