# Peer review of "The Overlooked and the Overstudied: A Scoping Review of Qualitative Research on Pursuing Sexual, Romantic, and Loving Relationships Through Online Dating"

_behavsci, 2025, doi:10.3390/bs15030247_

Round 1

Reviewer 1 Report

Comments and Suggestions for Authors

This is a meticulous research that examines, through a comprehensive scoping review of 125 qualitative studies on online dating, several patterns underlying the academic literature on this topic. It is well-written (although a thorough proofreading is still needed, as there are some annoying typos and grammatical errors) and the argument follows a logical structure for this type of material (i.e., scoping review). The most impressive feature of this research is certainly its scope, consisting of 125 academic articles published in several datasets in English. Consequently, I believe that this piece is suitable for publication in Behavioral Sciences, pending however addressing the series of issues detailed below.

11. Introduction (page 2, line 60-62): It is not clear why the authors added the sentence “Second, research serves as the foundation for further scholarly inquiry…”. This is a truism, and should not be stated, as every body of research is characterized by stimulating further inquiry (which is a core feature of scientific research actually). I suggest authors to drop this second point from the enumeration.

22. Research questions/objectives: The authors mention two “objectives and research questions,” which are actually formulated as research questions (and not as objectives). Please either add proper formulations for research objectives or revise the description.

33. Research questions: The authors specify two research questions. However, the first research question conflates demographic aspects with research methods. These are obviously distinct aspects and therefore belong to separate categories. I therefore suggest splitting them into distinct research questions (the paper will then have three research questions).

44. The last paragraph from “Introduction” (page 3, lines 98-103) should not be placed there, as its place is in the methodological section. It also repeats the immediately subsequent text from the following paragraph.

55. Previous literature reviews on online dating: The paragraph from page 3, line 128-135 confuses gender (male, female, non-binary) with sexual orientation (heterosexual, homosexual, etc.). Please address this issue.

66. Page 3, line 137: Please provide the full name for the acronym MSM when first mentioning the term.

77. Page 6, line 250: Please correct the typo in “Exclusion Catergories”.

88. Results/Age (page 7, lines 281-286): The authors refer to Generation Z, Millennials, Generation X, Baby Boomers, and the Silent Generation, with each cohort being characterized by some specific period. I would like the authors to be aware that these generational categorizations are based on the demographics and political/historical context of the United States in particular, and they have no correspondent in the Global South (or the post/Soviet societies, for that matter). As one of the key finding of this paper is that research tends to be concentrated on the Global North, and as author rightly argue for a broader geographical coverage, it is surprising to find that the author resort to such an US-centric generational categorization and extend it to the entire world.

99. Section 4.1.6. Qualitative Research Methods (page 8, line 332): Please consider discussing this as a separate independent section.

110. Page 11, line 457: Please consider deleting the sentence “We will now discuss our findings and conclude.” It is short, underdeveloped, and it should not feature as an entire paragraph.

111. Discussion and conclusions (page 12, line 519-520): “The last demographic category we explored were the different methodological tools used in qualitative research on online dating.” Please see point 9: methodology cannot be conceived as a “demographic category.” This section on methodological approaches should be a separate, independent section.

112. Page 14, lines 608-612: This entire paragraph is repetitive. I suggest the authors to consider deleting it to avoid unnecessary repetition.

113. Limitations (page 14, line 621): The authors acknowledge that one of the limitations of their research is the language (English only). I encourage them to develop the full implications of this limitation, by connecting the “English only” methodological decision to the overrepresentation of the “Global North” they found in carrying out this research.

114. Limitations (page 14, line 625): “A key limitation of our review is the sheer volume of available information.” The “sheer volume” cannot be a proper limitation. Perhaps the lack of tools or capacity to process it is the limitation. Please clarify and rephrase.

Comments on the Quality of English Language

The English is good. There are some typos and formulations that need to be addressed.

Author Response

Reviewer 1’s comment:

This is a meticulous research that examines, through a comprehensive scoping review of 125 qualitative studies on online dating, several patterns underlying the academic literature on this topic. It is well-written (although a thorough proofreading is still needed, as there are some annoying typos and grammatical errors) and the argument follows a logical structure for this type of material (i.e., scoping review). The most impressive feature of this research is certainly its scope, consisting of 125 academic articles published in several datasets in English. Consequently, I believe that this piece is suitable for publication in Behavioral Sciences, pending however addressing the series of issues detailed below.

Thank you for your thorough and constructive feedback. We appreciate your positive assessment and have carefully addressed the issues you highlighted, including proofreading for typos and grammatical errors. We have now revised the manuscript accordingly and look forward to your further evaluation.

  1. Introduction (page 2, line 60-62): It is not clear why the authors added the sentence “Second, research serves as the foundation for further scholarly inquiry…”. This is a truism, and should not be stated, as every body of research is characterized by stimulating further inquiry (which is a core feature of scientific research actually). I suggest authors to drop this second point from the enumeration.

Agreed, corrected.

  1. Research questions/objectives: The authors mention two “objectives and research questions,” which are actually formulated as research questions (and not as objectives). Please either add proper formulations for research objectives or revise the description.

Agreed, corrected.

  1. Research questions: The authors specify two research questions. However, the first research question conflates demographic aspects with research methods. These are obviously distinct aspects and therefore belong to separate categories. I therefore suggest splitting them into distinct research questions (the paper will then have three research questions).

Although we appreciate and understand the reviewer’s suggestion to separate RQ1 into two questions, we have chosen to keep it as one. This allows us to examine not only who is being studied but also how they are being studied, providing deeper insight into the intersection of potential gaps, biases, and trends in the field. We believe that separating them could weaken this connection and oversimplify the analysis. We have, however, added this information under the presentation of the research questions as well.

  1. The last paragraph from “Introduction” (page 3, lines 98-103) should not be placed there, as its place is in the methodological section. It also repeats the immediately subsequent text from the following paragraph.

Agreed, corrected.

  1. Previous literature reviews on online dating: The paragraph from page 3, line 128-135 confuses gender (male, female, non-binary) with sexual orientation (heterosexual, homosexual, etc.). Please address this issue.

Agreed, corrected.

  1. Page 3, line 137: Please provide the full name for the acronym MSM when first mentioning the term.

The full name of the acronym MSM is provided in the paragraph above, as the acronym is first introduced there. Therefore, the correction has not been made.

  1. Page 6, line 250: Please correct the typo in “Exclusion Catergories”.

Thank you, corrected.

  1. Results/Age (page 7, lines 281-286): The authors refer to Generation Z, Millennials, Generation X, Baby Boomers, and the Silent Generation, with each cohort being characterized by some specific period. I would like the authors to be aware that these generational categorizations are based on the demographics and political/historical context of the United States in particular, and they have no correspondent in the Global South (or the post/Soviet societies, for that matter). As one of the key finding of this paper is that research tends to be concentrated on the Global North, and as author rightly argue for a broader geographical coverage, it is surprising to find that the author resort to such an US-centric generational categorization and extend it to the entire world.

We appreciate this comment and have added a clearer explanation of why we have chosen to do it this way. Please let us know if it needs further elaboration.

  1. Section 4.1.6. Qualitative Research Methods (page 8, line 332): Please consider discussing this as a separate independent section.

We have chosen to keep it as one section but have elaborated on why in the text. See our answer to point 3.

  1. Page 11, line 457: Please consider deleting the sentence “We will now discuss our findings and conclude.” It is short, underdeveloped, and it should not feature as an entire paragraph.

Agreed, deleted.

  1. Discussion and conclusions (page 12, line 519-520): “The last demographic category we explored were the different methodological tools used in qualitative research on online dating.” Please see point 9: methodology cannot be conceived as a “demographic category.” This section on methodological approaches should be a separate, independent section.

Corrected. See our answer to point 3 and 9.

  1. Page 14, lines 608-612: This entire paragraph is repetitive. I suggest the authors to consider deleting it to avoid unnecessary repetition.

Agreed, deleted.

  1. Limitations (page 14, line 621): The authors acknowledge that one of the limitations of their research is the language (English only). I encourage them to develop the full implications of this limitation, by connecting the “English only” methodological decision to the overrepresentation of the “Global North” they found in carrying out this research.

Agreed, clarified.

  1. Limitations (page 14, line 625): “A key limitation of our review is the sheer volume of available information.” The “sheer volume” cannot be a proper limitation. Perhaps the lack of tools or capacity to process it is the limitation. Please clarify and rephrase.

We have deleted the sentence and the one following it, as we agree that this is not, in itself, a limitation.

We thank you very much for your thorough respons and hope our answers and revisions are satisfactory. 

Reviewer 2 Report

Comments and Suggestions for Authors

This is an interesting contribution to the field and this journal. I do suggest edits for clarification and improvement. 

Introduction/Review of Literature

·         Page 3, line 125: Motivations change during age? I’m unsure how motivations increase.

·         Page 3, line 135: Citation outside of parentheses.

·         Did the other reviews examine quantitative, qualitative, or both types of studies?

·         Page 4, line 150: What do you mean by “lacks coherence”?

·         Page 4, line 162: Self-report has been seen as very reliable. Why is not true here? Need a citation and explanation.

Methods/Results

·         Page 4, line 177: Is the last sentence missing a word?

·         Page 6, line 262: 335 for screening, not used in the study. It currently reads as if you have 335 articles in the analysis.

·         Very thorough and impressive methodology.

·         Section 1.4.6: More detail here please. For interviews and focus groups, what was the average number of participants? Because this study is uniquely focused on qualitative methods, more information should be provided in this section.

o   Perhaps in table A1, you could add a column for method and sample size.

·         How did you code the articles into themes? Please provide a thorough overview.

·         Page 10, line 384: Define ghosting.

·         Page 10, line 407: Hypertextualization or hyperSexualization?

Discussion

·         A consideration: Dating app studies skew younger because many of the apps are specifically targeted to younger adults. Tinder and Bumble advertised by going to Greek houses on campuses.

·         Page 12, lines 500-502: Did any of the studies look at rural/remote locations?

·         The paragraphs where you recap the sub-themes results can be significantly cut or reconceptualized. What do these results mean for future research in this area? Most of what you currently have in lines 557-590 have already been stated in the results section.

Comments on the Quality of English Language

Minor issues throughout that could be improved with thorough proofreading.

Author Response

Thank you for your feedback and valuable suggestions. We have carefully addressed the suggested edits for clarification and improvement in the revised version. We are now submitting the updated manuscript for your further consideration.

Introduction/Review of Literature

  • Page 3, line 125: Motivations change during age? I’m unsure how motivations increase.

Agreed. The sentence has been clarified.

  • Page 3, line 135: Citation outside of parentheses.

Thank you, corrected.

  • Did the other reviews examine quantitative, qualitative, or both types of studies?

We have added information regarding the methods of the studies included – though not for Wu and Ward nor Wu and Trottier, as they did not provide such information – as explained in the first paragraph under 2. Previous Literature Reviews

  • Page 4, line 150: What do you mean by “lacks coherence”?

Thank you. The statement is now better explained.

  • Page 4, line 162: Self-report has been seen as very reliable. Why is not true here? Need a citation and explanation.

The sentence on unreliability has been removed, as it is not a key priority for our topic and would require more space than we have available.

Methods/Results

  • Page 4, line 177: Is the last sentence missing a word?

We cannot see why this sentence is missing a word.

  • Page 6, line 262: 335 for screening, not used in the study. It currently reads as if you have335 articles in the analysis.

Thank you for addressing this. It appears that a part of the methods section was lacking, we apologize for that. We have added the requested information.

  • Very thorough and impressive methodology.

Thank you.

  • Section 1.4.6: More detail here please. For interviews and focus groups, what was the average number of participants? Because this study is uniquely focused on qualitative methods, more information should be provided in this section.

The requested information has been added.

o Perhaps in table A1, you could add a column for method and sample size.

While providing detailed methodology and sample size information in table A1 would be valuable, it would significantly exceed the word limit. However, we have addressed the reviewer's request by incorporating additional relevant details in the text.

  • How did you code the articles into themes? Please provide a thorough overview.

Thank you for noticing. A section on data analysis is added under 3.5.

  • Page 10, line 384: Define ghosting.

Definition added.

  • Page 10, line 407: Hypertextualization or hyperSexualization?

HyperSexualization – Corrected.

Discussion

  • A consideration: Dating app studies skew younger because many of the apps are specifically targeted to younger adults. Tinder and Bumble advertised by going to Greek houses on campuses.

Considered and added to the introduction (line 50-54).

  • Page 12, lines 500-502: Did any of the studies look at rural/remote locations?

As our knowledge on this aspect is unfortunately limited, we have decided to remove this sentence.

  • The paragraphs where you recap the sub-themes results can be significantly cut or reconceptualized. What do these results mean for future research in this area? Most of what you currently have in lines 557-590 have already been stated in the results section.

Thank you for noticing. The paragraphs have been re-written.

Comments on the Quality of English Language

Minor issues throughout that could be improved with thorough proofreading.

We have conducted a thorough proofreading of the manuscript and believe it now meets the required standards. We thank you and hope this revision is satisfactory.

Round 2

Reviewer 1 Report

Comments and Suggestions for Authors

The authors have made revisions to the originally submitted manuscript that addressed some of the issues pointed out in the review report. However, they have chosen to ignore the more substantial issues (e.g., the US-centric typology of generations that does not correspond to other places). Nevertheless, if the authors wish to publish it as such, as a reviewer I do not have something against it.